light microscopy/neuroscience

microscopy, ctenophore, electrophysiology, open hardware, epifluorescence, automation

**Author for correspondence:**
Amy Courtney
e-mail: amy.courtney@ucdconnect.ie

# The Flexiscope: a low cost, flexible, convertible and modular microscope with automated scanning and micromanipulation

Amy Courtney[1,2], Luke M. Alvey[1,2],
George O. T. Merces[1,2], Niamh Burke[1,2]
and Mark Pickering[1,2]

[1]School of Medicine, University College Dublin, Ireland
[2]UCD Centre for Biomedical Engineering, University College Dublin, Ireland

AC, 0000-0003-0495-1711; LMA, 0000-0003-4069-6383;
GOTM, 0000-0001-7116-2451; NB, 0000-0001-9293-6874;
MP, 0000-0002-2454-2897

With technologies rapidly evolving, many research institutions are now opting to invest in costly, high-quality, specialized microscopes which are shared by many researchers. As a consequence, the user may not have the ability to adapt a microscope to their specific needs and limitations in experimental design are introduced. A flexible work-horse microscopy system is a valuable tool in any laboratory to meet the diverse needs of a research team and promote innovation in experimental design. We have developed the Flexiscope; a multi-functional, adaptable, efficient and high-performance microscopy/electrophysiology system for everyday applications in a neurobiology laboratory. The core optical components are relatively constant in the three configurations described here: an upright configuration, an inverted configuration and an upright/electrophysiology configuration. We have provided a comprehensive description of the Flexiscope. We show that this method is capable of oblique infrared illumination imaging, multi-channel fluorescent imaging and automated three-dimensional scanning of larger specimens. Image quality is conserved across the three configurations of the microscope, and conversion between configurations is possible quickly and easily, while the motion control system can be repurposed to allow sub-micrometre computer-controlled micromanipulation. The Flexiscope provides similar performance and usability to commercially available systems. However, as it can be easily reconfigured for multiple roles, it can remove the need to

purchase multiple microscopes, giving significant cost savings. The modular reconfigurable nature allows the user to customize the system to their specific needs and adapt/upgrade the system as challenges arise, without requiring specialized technical skills.

## 1. Introduction

Microscopy is widely regarded as a centrally important technique in all areas of biological research. The ability to resolve structures which would have otherwise been invisible to our eyes has contributed to the advancement of many fields. For example, the field of neuroscience was revolutionized by the microscopic identification and description of neurons by Golgi and Cajal. Neuroscience today has been propelled by the advancement of microscopes to perform functional measurements and connectomic studies [1].

With technologies rapidly evolving, many research institutions are now opting to invest in high-cost, high-quality, specialized microscopes which are shared by many researchers. As a consequence, an individual user may not have the ability to adapt a microscope to their specific needs and limitations in experimental design are introduced from the outset. Each experimental design has an optimal opto-mechanical configuration, and the possible variations are diverse. For example, electrophysiology experiments frequently require a fixed specimen stage and motion control of the microscope. By contrast, fluorescent or differential interference contrast (DIC) microscopy usually requires motion control of the specimen stage and a fixed microscope. Depending on the sample, an upright or inverted objective orientation may be optimal. Many experiments also involve coupling a microscope with other equipment such as a micromanipulator or incubator. Multiple microscopes would be required for each application despite the fact that many elements would be duplicated across the configurations. An adaptable work-horse microscopy system is a crucial and invaluable tool in any laboratory to meet the diverse needs of a research team and promote innovation in experimental design.

With flexibility and funding limitations in mind, many researchers have devised excellent cost-saving strategies which include 3D-printed microscopes and *XYZ* translators [2–5], a $0.58 origami microscope [6] and modifications to old microscopes [5,7,8]. These types of systems are advantageous in the field or within incubators but their low cost often equates to a compromise in image quality and a lack of long-term stability. In addition, many laboratories do not have access to these salvaged components or 3D-printers and therefore reproducibility can be challenging. The components used to construct a flexible microscopy system must be readily available to research groups around the world, the system must cost considerably less than a commercial system while also maintaining a comparable image quality.

We have designed, constructed and extensively tested a high-quality, transformable microscopy system assembled from optical and mechanical components. The core optical components remain constant while alterations are made for specialized experimental set-ups, including objective orientation and which components are fixed or translating. The use of commercially available opto-mechanical elements offers many advantages including ease of use, reliable alignment and reproducibility (both within and between laboratories) due to the availability and compatibility of the parts. The modular, reconfigurable nature allows the user to customize the system to their specific needs and adapt the system to the everyday diverse challenges faced when attempting to answer complex neurobiological questions. This system can also be expanded to cope with new experimental challenges and upgraded as technologies rapidly evolve, a considerable advantage over static commercial systems. The flexibility of our modular microscopy system allowed us to also implement automated stage scanning and image acquisition. A commercial microscope would often require expensive and manufacturer-specific control software. Further cost savings are achieved as many components can be designated for multiple applications, such as the actuators used to control the automated specimen stage which can be easily reconfigured as a micromanipulator.

We have provided a comprehensive description of a multi-functional, custom-built, adaptable, efficient and high-performance microscopy/electrophysiology system for both standard and unconventional applications in a neurobiology laboratory. This system encompasses the capabilities of multiple microscopes with considerable cost and space savings. Our system can be directly replicated or adapted to suit the needs of any research group. Key characteristics of 'The Flexiscope' include: ease of use; upright and inverted configurations allowing multi-angle imaging; fluorescent microscopy; automated stage scanning; visualization of unstained tissue and sub-micrometre computer-controlled micromanipulation. Implementation of this system does not require any specialized skills or knowledge.

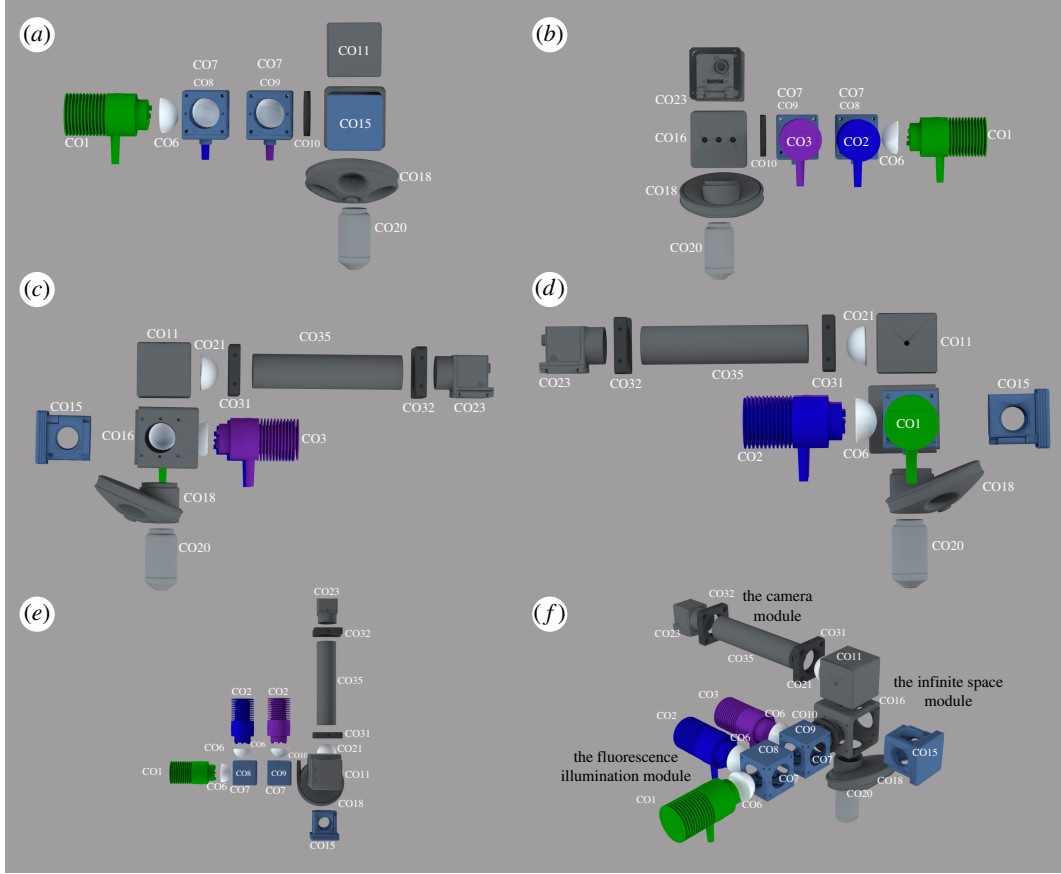

**Figure 1.** Core optical components of the Flexiscope. The key optical and mechanical components constituting the light path of the Flexiscope illustrated schematically. The core optical components remain relatively constant with each configuration. Depending on the application, alterations are made to the mounting and motion control of specific elements. (*a–d*) Illustrates the four side views of the core optical components of the Flexiscope. (*e*) Illustrates a top view of the core optical components of the Flexiscope. (*f*) Illustrates an orthogonal view of the core optical components of the Flexiscope. The system can be divided into three functional modules: the fluorescence illumination module, the infinite space module and the camera module. Component labels correspond to electronic supplementary material, table S1.

# 2. Material and methods

## 2.1. Parts and components

In order to thoroughly detail every component of the Flexiscope, each individual part has been allocated a part designation for use throughout this description. Electronic supplementary material, table S1 lists each part and its designation, in addition to the suppliers, supplier part number and cost of each part. Electronic supplementary material, file S1 details assembly/calibration instructions for the entire Flexiscope system.

### 2.1.1. The core optical components

We describe three sample configurations of the Flexiscope, differing in objective orientation, mounting and motion control. Within each configuration, the core optical components remain relatively constant (figure 1). The core optical components can be divided into three functional units: the fluorescence illumination module, the infinite space module and the camera module.

### 2.1.2. The fluorescence illumination module

Fluorescent excitation is provided by three independent high-intensity LEDs collimated with aspheric condenser lenses (CO6). They are combined into a single beam (figure 1*e,f*) using two dichroic mirrors

(CO8 and CO9) positioned at 45° to the light path in a mounting cube (CO7). The dichroic mirrors have 452–490 nm reflection/505–800 nm transmission and 360–407 nm reflection/425–575 nm transmission bands, respectively. This results in the 565 nm LED (CO1) passing through both dichroics, the 470 nm (CO2) reflecting off the first dichroic and passing through the second dichroic while the 395 nm (CO3) LED reflects off the second dichroic. An iris (CO10) is placed after the LED module before the filter mount-holding cube (CO16) to allow for regulation of excitation light. The excitation light is then directed towards the filter mount-holding cube (CO16). The filter mount-holding cube holds the swappable filter set mounts (CO15), which are pre-configured to contain an excitation, emission and dichroic filter corresponding to either 630 nm (Texas Red, CO12), 530 nm (FITC, CO13) or 460 nm (BFP, CO14) emission wavelengths. When the light passes through the excitation filter, the dichroic mirror inside the filter set mount directs the light towards the objective lens. The fluorescence illumination module is a fixed entity, regardless of the desired application.

### 2.1.3. The infinite space module

The Flexiscope is designed to use infinite conjugate objectives. The space between the tube lens and the objective (the infinite space module, figure 1*f*) can be modified to the user's needs with minimal impact on optical performance. Two infinite conjugate objectives, a 4× Olympus plan achromat objective (numerical aperture (NA) = 0.10, CO19) and a 20× Olympus water immersion objective (NA = 0.5, CO20), are coupled to a lens turret (CO18). The light path then travels vertically from the specimen, through the objective, through the filter set mount emission filter to the 45° turning mirror (CO11). The mirror directs the light path horizontally towards the achromatic doublet (CO21), which acts as a tube lens.

### 2.1.4. The camera module

The elements found immediately after the achromatic doublet lens compose the camera module (figure 1*e,f*). The distance between the achromatic doublet lens and the camera sensor is fixed at 150 mm using adjustable length lens tubes (CO28, CO35 and CO37). Transmitted or reflected light imaging can be achieved by leaving the filter mount-holding cube empty and enclosing the opening with the filter cube blank top plate (CO17). Oblique infrared illumination microscopy (OIR) uses infrared LEDs (CO26) at an angle above or below the specimen which allows the visualization of three-dimensional structures. We used a ring-type array of 36 850 nm LEDs; this type of array is designed for CCTV applications and is powered by a 12V DC supply. The array can be mounted on adjustable angle posts (MO6, MO7), in turn mounted in a height adjustable post holder (MO11). This arrangement allows oblique illumination with a wide, diffuse beam of the sample from any desired angle.

OIR has been used previously as a contrasting technique to resolve single cell structures in live, unstained tissue such as brain slice and spinal cord [9,10]. Direct comparison of this technique with DIC imaging has shown similar performance without the need of the specialized optics required for DIC [10], and the successful application of this simpler, more accessible technique led to our adoption of OIR in the Flexiscope. All cameras used are c-mount machine vision cameras from Point Grey (now owned by FLIR). A high frame rate, mono, infrared-sensitive camera is used for OIR (CO22). A high-sensitivity, low-noise, colour camera is used for fluorescence detection (CO23). The entire system is coupled together using 1 inch diameter lens tubes, with c-mount adaptors (CO24) used to couple the c-mount cameras.

## 2.2. Motorized motion control and automated image acquisition

Our motion control system was designed to allow precise, repeatable and controllable movement of the specimen stage. The system can be controlled using customizable Matlab and Python scripts and linked to image acquisition to permit fully automated scanning of larger samples.

### 2.2.1. The piezoelectric stage

Piezoelectric actuators (MC2) with 13 mm of travel control the three dimensions of motion of the specimen stage while the Flexiscope is in the upright or inverted configuration. This allows *X*-motion (stage: right/left), *Y*-motion (stage: forward/backward) and *Z*-motion (stage: up/down) control. A controller module (MC3) allows computer control of the actuators. The APT software (ThorLabs) allows control of the actuators through the controller module. The ActiveX control capability in

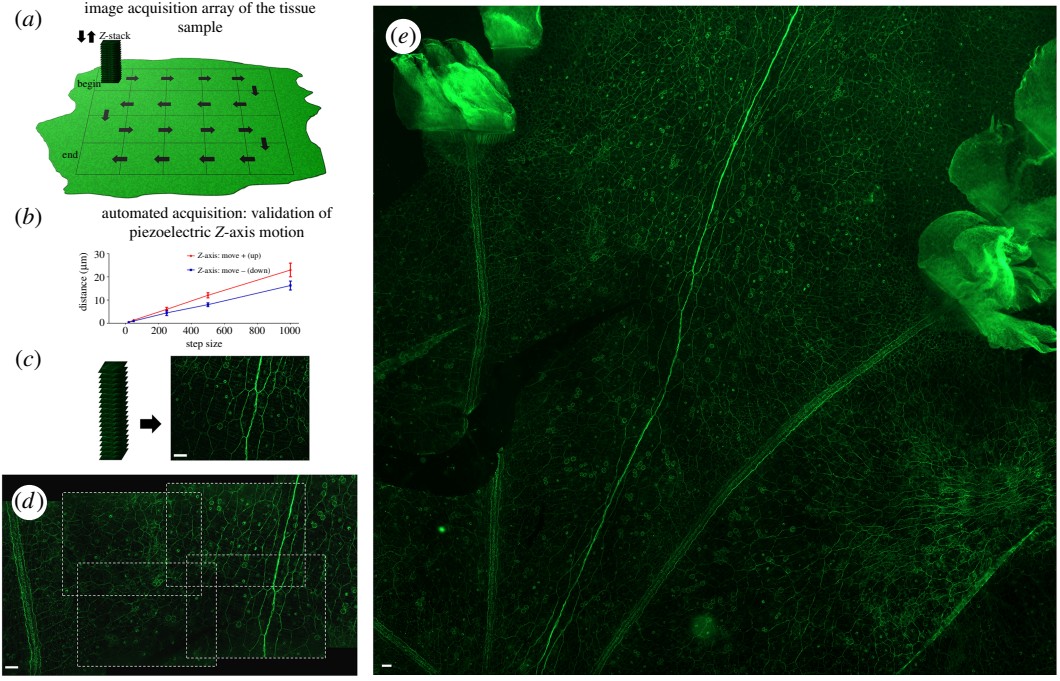

**Figure 2.** Validation of automated specimen stage motion control and image acquisition. (*a*) Schematic of the automated stage motion control workflow to acquire *Z*-stacks at regions of the sample in the *X* and *Y* directions. Many factors can influence piezoelectric actuator function, and as the 'step size' values are arbitrary, we acquired video recordings of the stage during motion and manually calculated the distance in ImageJ [11]. The following parameters were also set through the APT software; voltage (V): 110, drive rate (steps s$^{-1}$): 500, drive acceleration (steps s$^{-2}$): 100 000 and jog rate (steps s$^{-1}$): 500. Load can alter piezoelectric actuator function and thus the load of the *Z*-axis is approximately 335 g. (*b*) Validation of mean distance ($\mu$m) travelled by the *Z*-axis piezoelectric actuator when specific 'step size' commands (20, 50, 250, 500 and 1000) are executed repeatedly (*n* = 15). The error bars indicate standard deviation and thus the intrinsic lack of repeatability of these actuators is evident. The resolution limit did not enable individual step distance discrimination for the 20 and 50 'step sizes' and therefore standard deviation could not be acquired. The distance travelled when the stage is moving up is greater than the distance travelled when the stage is moving down, despite the same command being sent to the actuators. (*c*) Example of 20 images (*Z*-stack) acquired at each step in the *Z*-dimension to acquire multiple in-focus planes in the tissue sample. The corresponding maximum intensity projection is seen alongside the *Z*-stack. This allows the three-dimensional tissue to be visualized as a two-dimensional image [12]. (*d*) Illustrates the overlap in each new FOV in the *X* and *Y* dimensions and enables image stitching. (*e*) A composite of 50 maximum intensity projection images stitched together in ImageJ [13]. Raw data corresponding to (*b*) can be found in electronic supplementary material, file S2. Scale bar for all: 50 µm.

Matlab allows control of the actuators via the APT software. The image acquisition toolbox in Matlab is also capable of controlling the PointGrey cameras used on our microscope, so automated scanning is operated within Matlab's workspace.

The commands used to control the specimen stage followed a logic in which the image sequence of the tissue sample is considered a two-dimensional array (figure 2*a*). To acquire images of the whole tissue (or a region of interest), the stage is moved in the *X*-axis *n*-times in one direction, then moved in the *Y*-axis once and in the opposite direction in the *X*-axis *n*-times again. This cycle can then be repeated until the tissue has been imaged in its entirety. Each movement in *X* or *Y* dimension reveals a new 'field of view' (FOV) and a 'Z-stack' is subsequently acquired. A Z-stack is a series of movements in the *Z*-dimension while images are acquired. The number of images acquired is defined by the user and depends on the thickness of the tissue (for code and detailed user guidelines see electronic supplementary material, file S1).

### 2.2.2. The stepper motor automated scanning stage

While the piezoelectric automated scanning stage was effective for our application (see §3.2), one major limitation became evident during extensive use; the piezoelectric actuators were slow. Another key

consideration is that the while these actuators operate with a typical step size of 20 nm, the actual displacement is heavily dependent on load and torque, and so absolute displacement in a motion control system based on piezo actuators is difficult to predict reliably (see characterization in §3.2.1), and needs to be calibrated for the load. We therefore decided to reconfigure the automated scanning stage to incorporate stepper motors (electronic supplementary material, figure S1). This stage used 3D-printed components to transform a manual $XYZ$ translating stage with standard micrometers (MC4) into a motorized stage. These micrometers had the same 13 mm travel range as the piezo actuators. The micrometer on each axis was coupled directly to the axis of a stepper motor using 3D-printed coupler (3D3). To ensure that the stepper motor itself did not rotate, it was mounted into a mounting plate (S11), which was coupled to the translating stage using 10 mm MakerBeam aluminium extrusions. This kept the motors aligned to the planes of translation during movement. In the $Y$ and $Z$ axes, the motor mounts were coupled to the actual translating part directly using 3D-printed parts (3D4, 3D5). This was not possible on the $X$-axis, so here the motor mount was attached to a MakerBeam frame, and bearings (S10) attached to the corner of the frame and aligned to two MakerBeam extrusions (S15), perpendicular to the motor frame and coupled to the base (MO15). This configuration allowed the motor to translate with the moving stage, while constraining the motor from rotating, as the frame slid along the S15 beams using the bearings. The arrangement of all mounting components is shown in electronic supplementary material, figure S1. One full rotation of the micrometer will advance the actuator by 500 µm. As the stepper motors rotate 1.8 degrees/step (i.e. 200 steps/revolution), and they are directly coupled to the actuator, each step corresponds to 2.5 µm displacement of the stage. However, stepper motors can be moved in fractional microsteps. A 1/16 microstep then will produce a translation of 156.25 nm microstep$^{-1}$. While this is significantly more than the step size of the piezo stepper system, it should be more predictable and less affected by load.

Stepper motors are controlled using an Arduino Uno running a gcode interpreter. Gcode is a standardized system for controlling $XYZ$ position and movements commonly used in 3D-printers and CNC machines. Control of the stage can then be achieved by issuing gcode commands. The system is controlled in Matlab or Python in a manner similar to the piezoelectric configuration and the code/user guidelines are available in electronic supplementary material, file S1.

## 2.3. Applications of the Flexiscope: transforming between configurations

A key design principle of a transformable microscope like the Flexiscope is that a single user should be able to convert from one mode of operation to another in a relatively short space of time, and without requiring specialized skills or tools. The core optical components are relatively constant in the following three modes of operation: the upright configuration, the inverted configuration and the upright/electrophysiology configuration, but differ primarily in mounting and motion control.

### 2.3.1. Inverted to upright configuration in under 30 min

The steps required to transition from the inverted to upright configuration (figure 3) involve firstly removing the $XYZ$ specimen stage (MC4) from the jack (MC1). The jack and the microscope can then be removed from the optical table (MO19). The microscope is simply rotated 180° so that the objectives are no longer facing vertically up, but now facing vertically down towards the optical table. Post and post holder height on the microscope is increased (MO11, MO5, MO4). The microscopy slide holder (MO18, MO17, MO16) on the specimen stage is reconfigured as seen in figure 3f. The specimen stage is mounted on a breadboard (MO15) with posts (MO10), post holders (MO8) and mounting bases (MO12, MO13). The microscope can then be mounted on the jack. Finally, the specimen stage and microscope/jack are aligned to one another before fixing all components to the optical table. This process takes under 24 min (electronic supplementary material, video S1). Performing these steps in reverse will result in transformation from the upright to the inverted configuration.

### 2.3.2. The upright/electrophysiology configuration: manual microscope translation with a four-dimensional micromanipulator controlled by piezoelectric actuators

The third configuration (upright/electrophysiology configuration, figure 4) requires some alterations to core optical components. For ease of mounting, in this configuration the light path is aligned vertically. To achieve this, the 45° mirror in the infinite space module is replaced with a 1.5 inch straight lens tube

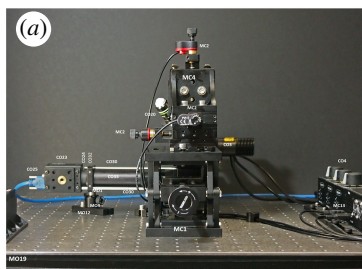
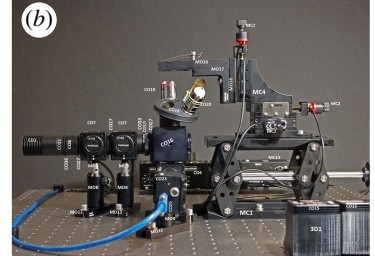
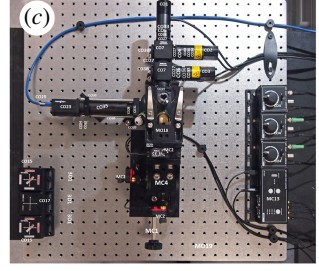

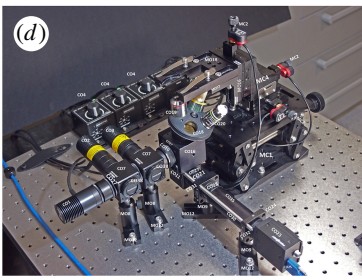

| key characteristics of the inverted configuration |
| --- |
| fixed microscope height |
| adjust specimen stage height with jack |
| precise *XYZ* control of specimen with 13 mm piezoelectric actuators |
| requires lightproof housing |
| fluorescence and IR-OI compatible |

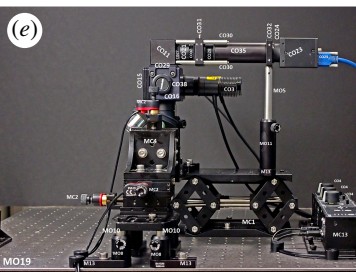
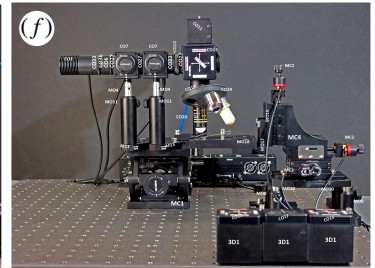
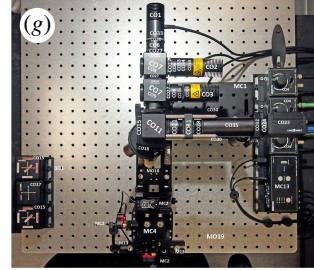

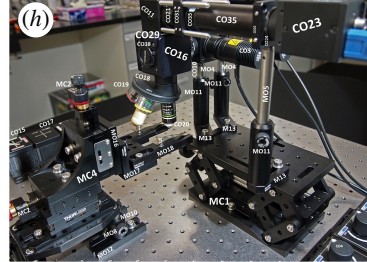

| key characteristics of the upright configuration |
| --- |
| ability to adjust microscope height with jack |
| precise *XYZ* control of specimen with 13 mm piezoelectric actuators |
| fluorescence and IR-OI compatible |

**Figure 3.** Comparison of the inverted and upright configurations of the Flexiscope. Photographs of the inverted (*a–d*) and upright (*e–h*) configurations illustrate the arrangement of the key optical and mechanical components of the Flexiscope into commonly used set-ups. Conversion between these configurations can be achieved in under 24 min (electronic supplemental material, video S1). Key features of each configuration are listed alongside the photographs. Component labels correspond to electronic supplementary material, table S1.

(CO39). In addition, the fluorescence illumination module is rotated so that the side-mounted LEDs are oriented upwards. This modification is only required to facilitate mounting to the optical breadboard (MO15). The whole microscope is mounted to this breadboard, which is in turn mounted vertically to a manual *XYZ* translation stage (MC6). The microscope components are mounted to the jack. A fixed specimen stage is mounted directly to the optical table.

Precise sub-micrometre micromanipulation of an electrode is imperative when undergoing electrophysiology experiments. However, such a tool can be extremely costly, especially when computer-controlled systems are required. The piezoelectric actuators which were previously configured as a specimen stage are now reconfigured as a micromanipulator (figure 5). A fourth dimension (*approach*-axis or *A*-axis, MC5) is incorporated into the stage and the angle of approach can be manually adjusted, depending on the objective configuration or to reduce the length of electrode in solution (minimize electrical noise). The electrode holder is mounted to the micromanipulator using Makerbeam and Thorlabs components (HS1–4). This configuration therefore involves a fixed specimen stage, manual microscope motion in three dimensions and four-axis micromanipulation with less than 1 μm step size (electronic supplementary material, video S2).

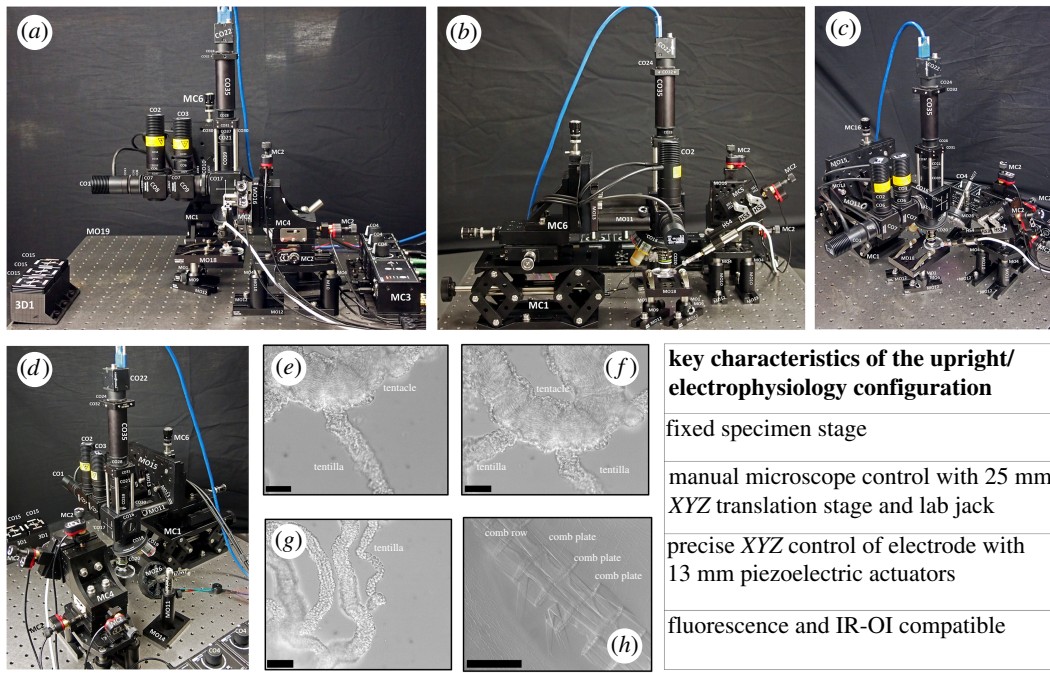

**Figure 4.** Configuring the Flexiscope for electrophysiology experiments and demonstrating OIR microscopy capabilities of the system to visualize unstained tissue. Photographs of the upright/electrophysiology configuration (*a–d*). Component labels correspond to electronic supplementary material, table S1. The key features of this configuration are listed alongside the photographs. The tentacles of *P. pileus* (gelatinous marine invertebrate) were used to demonstrate the capability of the Flexiscope in any configuration to implement OIR microscopy (*e–g*) scale bar: 50 µm. OIR microscopy to visualize the ciliary structures and body wall of *P. pileus* (*h*) *s*cale bar: 500 µm. OIR microscopy is performed by simply positioning infrared LEDs obliquely above or below the tissue. No staining was required and the three-dimensional architecture of the tissue could be appreciated.

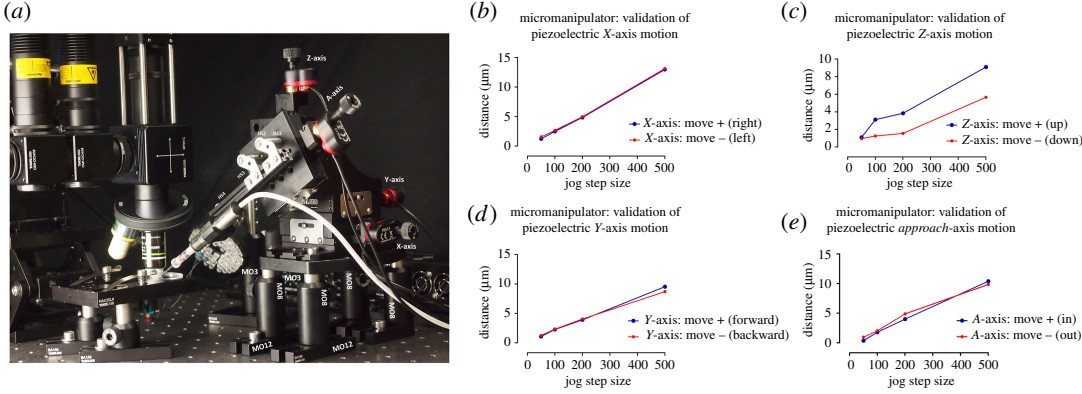

**Figure 5.** Micromanipulator validation: piezoelectric actuation in four dimensions with less than 1 µm resolution. Photograph of the *XYZ* stage and piezoelectric actuators configured as a micromanipulator. An additional axis known as the *approach*-axis (*A*-axis) is incorporated to the stage to allow four-dimensional computerized motion control (*a*). Graphs (*b–e*) outline the mean distance (µm) achieved when commands of a specific 'step size' were sent to the *X*, *Y*, *Z* and *A* axes piezoelectric actuators, respectively. Many factors can influence piezoelectric actuator function and as the 'step size' values are arbitrary, we acquired video recordings of the stage during motion and manually calculated the distance in ImageJ [11]. The following parameters were also set through the APT software for each piezoelectric actuator; voltage (V): 110, drive rate (steps s$^{-1}$): 500, drive acceleration (steps s$^{-2}$): 100 000 and jog rate (steps s$^{-1}$): 100. Load can alter piezoelectric actuator function and thus the load of each axis was calculated approximately; *X*-axis: 1147 g, *Y*-axis: 692 g, *Z*-axis: 435 g and *A*-axis: 85 g. Raw data corresponding to (*b–e*) can be found in electronic supplementary material, file S2.

## 2.4. Preparation of tissue samples for validation

### 2.4.1. *Pleurobrachia pileus* tissue preparation

The ctenophore *Pleurobrachia pileus* (a gelatinous marine invertebrate) was collected from Howth Harbour, Dublin, Ireland (53°23′35.4″ N 6°03′59.5″ W) between May and September 2017 using

custom-made plankton nets. Animals were maintained in a specialized aquarium system at 12–14°C in University College Dublin, Ireland for less than 24 h.

This animal possesses a decentralized neural network known as a 'nerve net' which is distributed across their spheroidal body, directly beneath the epithelial layer [14,15]. Whole mount immunofluorescence labelling of tyrosylated α-tubulin enables visualization of their nervous system. Tissue was processed in a manner similar to [15]. Animals were fixed using 50% natural seawater and 50% paraformaldehyde (PFA, 8%) in phosphate-buffered saline (PBS) (Thermo Scientific, 10 209 252) at room temperature. Depending on animal size (blotted live weight: 0.047–1.249 g), fixation occurred for 1–8 h. Animals were dehydrated with 50% ethanol (100%) and 50% PBS for 15 min at room temperature then 100% methanol for 15 min at room temperature and stored in methanol at −20°C.

When the tissue was ready to be processed for immunofluorescence experiments, rehydration to PBS was undertaken (the above dehydration steps in reverse). The tissue was dissected to enable 'flat' mounting of the outer epithelial layer to a glass slide. All internal anatomical structures (mesoglea, gastrovascular canals, tentacles, tentacular sheaths, mouth and pharynx) were teased away with Dumont HP Tweezers (5 Carbon steel 0.08 × 0.04 mm tip). The tissue was permeabilized with TritonX-100 (Sigma, Aldrich, T9284) at 0.2% and 0.1% in PBS for 30 min each on a rocking platform at room temperature. A fluorescent conjugated primary antibody against tyrosylated α-tubulin (Novus Biologicals, YL1/2, DyLight 488, NB600-506G) was diluted to 1:1000 in 1% BSA (Sigma-Aldrich, A7906) (diluted in 0.01% Triton-X100) and incubated in the fridge overnight. After washing the tissue in PBS three times for 10 min each, the tissue was placed epithelial layer up on a slide with OCT compound (VWR, 361603E) and sealed with a coverslip. This stabilizes the tissue and allows one- to two-week storage in a humidity chamber in the fridge. Fluorescence imaging in all three configurations was tested with this tissue preparation at the same position to allow direct comparison.

### 2.4.2. Dorsal root ganglia preparation

Five-day-old Wistar rat pups that were obtained from the Biomedical Facility in University College Dublin were euthanized in accordance with institute guidelines and relevant legislation (directive 2010/63/EU). Dorsal Root Ganglia (DRGs) dissected from the pups were cultured as explants for 6 days on flat laminin-coated silicone substrates and fixed in PFA. Axons were labelled with 1:1000 chicken anti-neurofilament heavy (NFH) primary antibody (ab72996, Abcam) and 1:500 goat anti-chicken IgY Alexa Fluor®568 secondary antibody (ab175477, Abcam). Cytoplasm of the cell bodies of migrating Schwann cells were labelled with 1:500 rabbit anti-S100β primary antibody (ab52642, Abcam), and 1:500 goat anti-rabbit IgG Alexa Fluor®488 secondary antibody (A11008, Fisher Scientific). Nuclei of Schwann cells were labelled with the DNA label 10 µg ml$^{-1}$ DAPI (D9542, Sigma).

## 2.5. Characterizing optical resolution and system mechanical stability

The resolution of the 20× objective (NA = 0.5, CO20) was investigated using a USAF 1951 resolution test target (CA1) in all configurations. This was performed in a manner similar to Cybulski et al. [6]. The Abbe diffraction limit of this lens is equal to the wavelength (calculated as $d = \lambda/2NA$ and NA = 0.5), and the Rayleigh limit is therefore the wavelength ×1.22 (e.g. 647 nm for 530 nm light). The smallest group and element on our test target is group 7 element 6. Line pairs mm$^{-1}$ of a given group and element is calculated using this equation; $RES_{LP} = 2^{Group+(Element-1)/6}$, and the distance between the lines in micrometres is equated using this formula; $RES_{CC} = 1000/RES_{LP}$.

Additionally, we characterized the optical performance of the whole system by estimating the slant-edge modulation transfer function (MTF), which is the basis for the ISO 12233 standard for testing camera systems [16]. This calculation was derived from an image of slanted edges of the solid blocks in the same USAF 1951 test target used above in the upright configuration, again acquired using the 20× objective.

To assess mechanical stability of the system over time, we used the same solid block on the test target, but this time focused on the corner. This gave an easily defined fixed point which could be used to measure drift over time. The corner was imaged every 10 s over 10 min, and then every 10 min over 4 h. The displacement of the corner pixel in the image from frame to frame in this time series was then used to measure drift. Again, the 20× objective was used, giving a pixel resolution of 0.386 µm pixel$^{-1}$. We carried out this test in the electrophysiology configuration, as this was likely to be the least mechanically stable (least number of mounting posts and all mounted horizontally) and therefore the 'worst-case scenario' for mechanical stability of the system.

# 3. Results

## 3.1. Validation of fluorescent and OIR microscopy

The use of whole mount prepared *P. pileus* tissue served as an appropriate stress test for our system as the complex three-dimensional topography of the tissue requires Z-stacking. Fluorescence imaging in all three configurations was tested with this tissue preparation at the same position to allow direct comparison (figure 6a–f). No apparent difference in the quality of images produced by the system was observed between the three configurations. The optical resolution was tested in all configurations using a USAF 1951 resolution test target (CA1) and the 20× objective (NA = 0.5, CO20). All configurations resolved group 7 element 6, which equates to a resolution of 228 line pairs mm$^{-1}$ or 1.1 µm. This is the smallest set of lines on our test target; therefore, the resolution of the system is probably less than this. No discernible difference in resolution is seen for each configuration (electronic supplementary material, figure S2A–I). The slant-edge MTF was calculated for both the horizontal and vertical edges. The horizontal and vertical MTF profiles are presented in electronic supplementary material, figure S3 (A and B). We also investigated the uniformity of illumination of the three LEDs. We found the illumination was relatively uniform and sufficient for fluorescent excitation across the FOV (electronic supplementary material, figure S2 J–L).

Additionally, the mechanical stability of the system over time was evaluated. Over 10 min, the corner pixel of the opaque square on the test target moved no more than one pixel in either X or Y (electronic supplementary material, figure S3c,d). Over a period of 4 h, some drift was evident along both axes (electronic supplementary material, figure S3e,f), although this was limited to 7 pixels (less than 3 µm) in any plane. Drift in Z wasn't measured directly; however, there was no clear evidence of a shift in focus over the 4 h period of imaging.

The triple fluorescent labelling of the DRGs allowed us to test the performance of the manually swappable filter cubes and assess if multi-channel imaging of a sample at the same location is possible. The system was capable of three-channel imaging (figure 6g–i) and subsequent alignment to produce a composite image (figure 6j). This demonstrates three-channel fluorescent capabilities of the system.

OIR was tested in the upright/electrophysiology configuration (figure 4e–h, electronic supplementary material, video S3). This technique allowed clear visualization of unstained *P. pileus* tissue and the three-dimensional topography of the tissue was discernible. This approach also allows for the clear visualization of the electrode tip, critical for electrophysiological applications (electronic supplementary material, video S2).

## 3.2. Validation of automated stage scanning and image acquisition

In order to assess the effectiveness of the automated specimen scanning stage, we wanted to satisfy four specific criteria: relatively low cost, reasonably fast speed of motion, repeatability of motion and multi-functionality (ability to also function as a micromanipulator).

### 3.2.1. The piezoelectric stage

#### 3.2.1.1. Repeatability in the Z-axis

At each new FOV, the Z-axis will move a specific 'step size' a defined number of times. This results in a Z-stack of images in which the three-dimensional qualities of the tissue can be appreciated as a maximum intensity projection [12] (figure 2c). The distance travelled by the piezoelectric actuators with each 'step size' is dependent on many factors, including the resistive torque against which the actuator tip is pushing, drive voltage, step rate, active preload, variance in the frictional behaviour of assembly components and actuation direction/condition [17]. The 'step size' values used by the APT software are arbitrary; therefore, we needed to determine the distance in µm these values equated to. We also wanted to test the repeatability of each step size as it is reported that the distance may vary up to ±20% with each step [18].

Videos were acquired of the stage in motion to determine the distance travelled for 'step sizes' of 1000, 500, 250, 50 and 20 (figure 2b). The mean distance (µm) over 15 executions of a step was calculated. The intrinsic lack of reliability upon repeating the same command was clearly observed during our testing as seen in figure 2b. This is not a major issue for our applications as maximum intensity projections disregard any Z-dimension information. However, repeatability in the Z-axis would enable three-dimensional reconstructions of the Z-stacks (for example, Fiji's Z-project [19]); therefore, a different motor should be used to achieve this. In addition, the distance travelled when

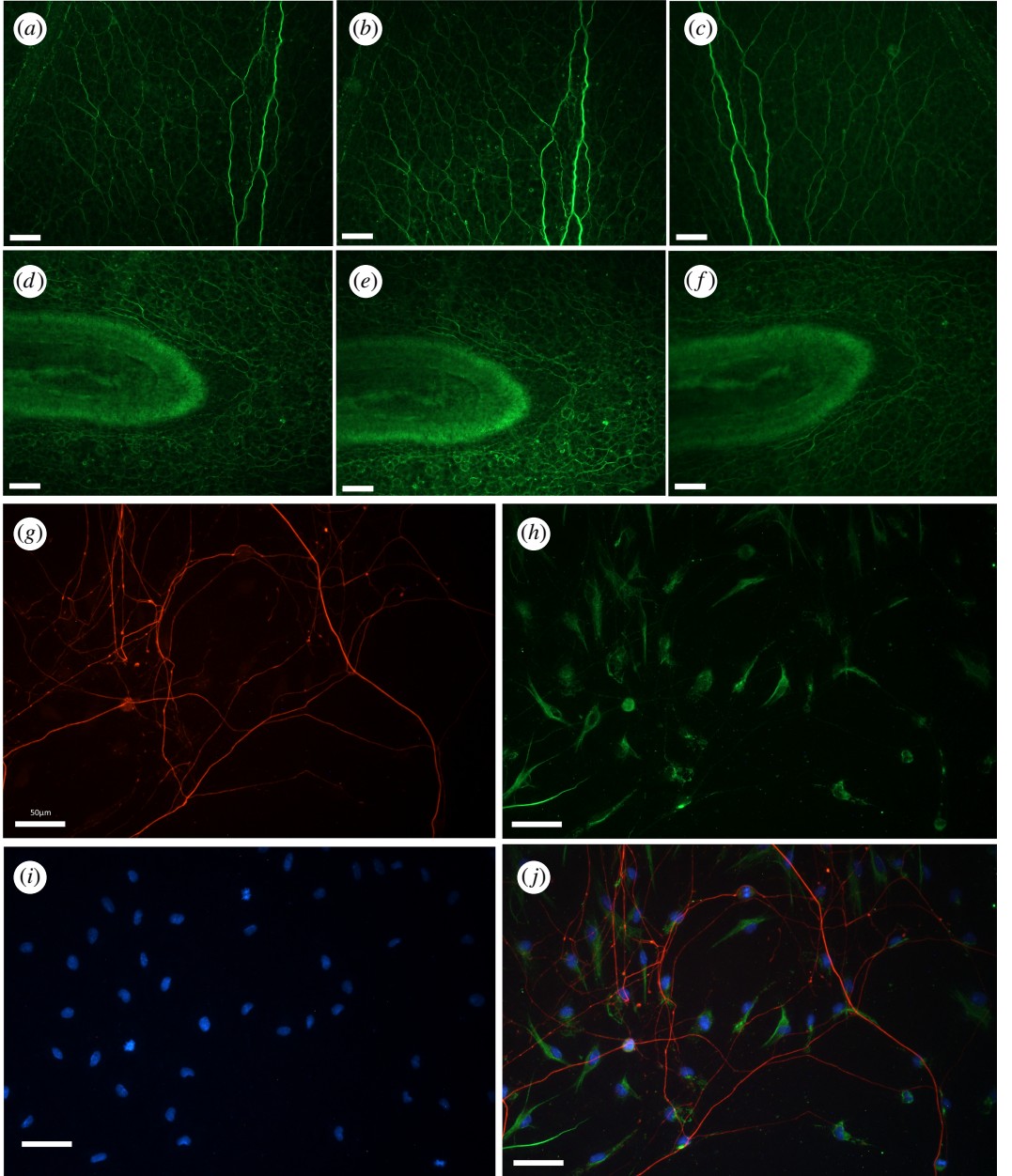

**Figure 6.** Comparison of fluorescent microscopy performance in all three major configurations and demonstration of three-channel fluorescent capabilities of the Flexiscope. Fixed whole mount *P. pileus* (gelatinous marine invertebrate) were labelled with an antibody against anti-tyrosylated α-tubulin. Images (*a*) and (*d*) were taken with the inverted Flexiscope configuration. Images (*b*) and (*e*) were taken with the upright Flexiscope configuration. Images (*c*) and (*f*) were taken with the upright/electrophysiology Flexiscope configuration. Images (*a–c*) and images (*d–f*) were acquired at the same region of tissue and thus allowed direct comparison of fluorescence performance in each configuration. The high-quality performance was consistent for each configuration. Images (*g–j*) demonstrate the three-channel fluorescent capabilities of the Flexiscope. Images (*g–j*) represent the same region of DRGs dissected from 5-day-old rat pups which were cultured as explants for 6 days on flat laminin-coated silicone substrates and fixed in PFA. Image (*g*) demonstrates NFH immunolabelling to visualize axonal outgrowth. Image (*h*) represents immunolabelling for S100$\beta$ to visualize cytoplasm of the cell bodies of migrating Schwann cells. Image (*i*) demonstrates DAPI (nuclei marker) staining. Image (*j*) is an overlay of Images (*g–i*). Scale bar for all images: 50 μm.

the stage is moving up is greater than the distance travelled when the stage is moving down, despite the same distance command being sent to the actuators.

### 3.2.1.2. Z-axis error correction

As mentioned in §3.1, imaging the structurally complex three-dimensional topography of *P. pileus* tissue endows many challenges including the fact that the focal plane of the tissue will vary throughout the

sample. This factor in combination with the discrepancy in distance travelled when the stage is moving up/down resulted in the need for the implementation of a Z-correction mechanism. A previously described auto-focus algorithm [20] was implemented on all the images in every Z-stack. This enabled a correction factor in the Z-axis to be implemented prior to the next Z-stack acquisition. Over time, without the use of this Z-correction, the Z-stack images would not contain any in-focus regions and thus this was an effective solution to this problem.

### 3.2.1.3. Large composite image generation

An example of a tissue scan composite is seen in figure 2*e* and demonstrates automated stage scanning capabilities. This demonstrates the nerve net in *P. pileus* as described in §3.1. This composite is a combination of 50 images acquired at 20× with each image comprising a maximum intensity projection from 20 images (generated using stitching algorithm ImageJ: [13]). This scanning capability not only enables significant time saving (as opposed to performing this manually), but it also enables an appreciation of information flow and overall context of the network which is otherwise lost with sub-sampling small regions of the nerve net.

### 3.2.1.4. Repeatability in the *X* and *Y* dimensions

The *X* and *Y* actuators were tested and optimized for 20× and 4× objectives to determine the appropriate travel distance to achieve a new 'FOV' with sufficient overlap for subsequent image stitching. The *X* and *Y* axes FOV overlap at 20× was established at 25% and 30%, respectively (figure 2*d*). This overlap value can be decreased or increased depending on the specific application/tissue by editing the distance command in the code. The lack of repeatability of the piezoelectric actuators was also observed in the *X* and *Y* axes; however, the overlap is large enough to overcome this inconsistency using feature matching stitching algorithms rather than an *XY* coordinate approach [13].

### 3.2.2. The stepper motor stage

The stepper motor stage aimed to improve the speed and repeatability of the piezoelectric configuration. The *X* and *Y* axes had a FOV overlap at 20× of 10%, and this was sufficient to enable subsequent image stitching. This system could translate to a new FOV in the *X*-dimension at 4× magnification in 15 s (as opposed to 290 s in the piezoelectric assembly). The speed that the piezo actuators can move at is limited by their design and the maximum step frequency that the driver can produce (the manufacturer-specified top speed is 3.5 mm min$^{-1}$, although we only achieved much slower speeds). Stepper-based systems can have much higher step rates (in kHz range), and with a full step size of 2.5 µm, they are inherently capable of much faster translation speeds. The stepper system assembly costs €1666 as opposed to €4539 for the piezoelectric system. In addition, Python and Matlab code are available to control the stepper stage, whereas only Matlab code was devised for the piezoelectric stage. This represents another cost saving. The discrepancy between actual distance travelled and the Gcode distance command was −0.56% and up to +36.35% (electronic supplementary material, figure S1). This was most likely due to a mechanical issue (3D-printed coupling components not perfectly aligned) rather than a motor or software issue. The repeatability of the movements was improved when compared with the piezoelectric actuator configuration. This was assessed by measuring the mean and standard deviation of the same movements repeated 10 times (electronic supplementary material, figure S1). We also characterized the stability of this stage during motion. Oscillations in the *X* and *Y* planes of the image were observed after motion had ceased. A 6 µm Z-axis movement (including residual oscillations) comes to a complete stop after 328 ± 22 ms. In the *X* and *Y* axes, the residual oscillations after the motor has stopped moving lasted 237 ± 149 ms and 355 ± 43 ms, respectively. Theses vibrations are not an issue for this system's functionality as an automated scanning stage, as image acquisition is never undertaken while the stage is in motion and we factored in a sufficient wait time before an image was acquired. These oscillations indicate that this configuration would not be suitable as a micromanipulator. The advantages of this system as compared with the piezoelectric system include cost and speed of imaging.

## 3.3. Validation of piezoelectric actuators as a micromanipulator in four dimensions

As described above, the same approach was taken to determine the distance (µm) travelled when commands of a specific 'step size' were sent to the piezoelectric actuators, now configured as a

micromanipulator (figure 5*a*), to ensure the effect of a different load was accounted for. Precise electrode control in four directions was achieved (figure 5, electronic supplementary material, video S2). Figure 5 outlines validation of the piezoelectric actuators as a micromanipulator for electrophysiological recordings. These actuators are described as being ideal for set and hold applications as they are self-locking and no power is required to hold position [18]. We therefore tested stability of the electrode position over time as this is crucial during intracellular electrophysiology experiments. At a resolution of 1.85 µm per pixel, no drift was observed over 16 h with the power off. The lack of repeatability is not a major issue for micromanipulation applications.

# 4. Discussion

We have provided a detailed description of the Flexiscope, a modular custom-built microscopy and electrophysiology system which is tailored to the specific needs of our research and successfully achieved substantial cost savings. This system can be controlled in Matlab or Python and could be replicated or adapted for the specialized needs of any researcher. The system is optically simple and constructed using simple screw-fit components that are readily commercially available for the most part. Given this, even a novice user with all required parts on hand should be able to construct the system in less than a day.

## 4.1. Piezoelectric stage versus stepper stage

In our case, an *XYZ* computer-controlled stage was used for automated stage scanning and micromanipulation. In theory, both functionalities can be performed with the same *XYZ* motion control system. However, the criteria that are critical for optimal operation are different in each mode. The stage scanning needs to be as fast as possible with relatively repeatable step sizes. Micromanipulation needs to have fine control at low resolutions and stability, whereas speed and repeatability are not a major issue. The piezoelectric stage was successful for stage scanning and micromanipulation. However, the motors were slow and not optimal for the stage scanning of larger samples (mm–cm scale). The stepper motor stage was devised to solve this problem of speed. An increase in speed and repeatability was noted with the stepper stage thus making it the superior option for stage scanning. Vibration during movement arose most likely due to misalignment of the stepper motors, which were mounted directly to the stage. However, this was not an issue as images are acquired after the stage has stopped moving. Overcoming this issue would enable this configuration to also be used for micromanipulation albeit with a higher resolution of approximately 2 µm as compared with sub-micrometre resolution for the piezoelectric actuators.

## 4.2. Cost breakdown

To enable fair comparison in terms of overall cost of our system, it is appropriate to mention the customizable Cerna Microscope available from Thorlabs. A system capable of fluorescence with microscope translation costs €49 000 ex VAT and does not come with objectives, filters or camera. Purchasing the entire Flexiscope as described here including the piezoelectric stage, the optical breadboard, blackout enclosure, two cameras and two objectives would amount to approximately €16 000. Further cost savings could be made by only having one channel of fluorescence which would bring the cost down to approximately €13 000. Purchasing the entire three-channel Flexiscope as above but switching to the stepper stage rather than the piezoelectric stage would cost approximately €13 000. The cost of the Flexiscope without any motion control elements is approximately €11 000. Depending on your requirements, other excellent motion control systems have been devised by Campbell *et al.* [21] and Sharkey *et al.* [2]. It should also be noted that while Matlab was available to all staff and students in our institution, this may not be the case in other institutions and this cost should be taken into consideration. Pricing is regionally specific, but the academic, single user, perpetual licence, including the required imaging toolboxes can be purchased in Ireland for €900, as listed in electronic supplementary material, table S1. To provide an alternative to this we also translated our code into Python to improve the accessibility of this tool and to further promote an open source ethos which this scope encompasses. The Python code uses the FLIR/PointGrey SDK and while the SDK is freely available to all of their customers, the user agreement prohibits us from sharing any code which uses their libraries. We have therefore provided a Python script with the FLIR/PointGrey code removed but

described to enable the reader to insert the appropriate commands upon purchasing FLIR products. Alternatively, the FLIR/PointGrey camera and motion control systems (using ThorLabs APT for piezos or Arduino control for the steppers) are all currently supported in µManager [22], so integrating this system into a µManager-based workflow should also be possible.

## 4.3. Comparison with other modular microscopes

There are few microscopy systems which can be directly compared with our system. In terms of the ability to alter the light path to configure the objective in the upright or inverted configuration, this has been described once previously [23]. Other modular epifluorescent [9], confocal [24] and scanning two-photon [25] microscopy systems have also been described. However, our system advantage lies in flexibility, specifically in the ability to designate multiple roles to specific components and alter what aspect of the system is fixed or translating in each configuration. This allows us to achieve the capabilities of multiple microscopes at a considerably lower cost.

## 4.4. Limitations

A small number of limitations of the system were noted over many months of extensive use. Dust can be easily introduced to the system during assembly and reconfiguration; even with care this seems inevitable, although deconstruction for cleaning should actually be easier than pre-assembled commercial systems. Wear/tear and potential damage due to the repeated handling of the various elements is a possibility but any damage to a specific element can be easily replaced at a relatively low cost. By contrast, replacing a specific element of a commercial microscope is expensive and often must be performed by an engineer. A degree of uneven illumination was observed in our fluorescent images; however, this has been described as a common problem in many commercial systems [26]. In addition, during motion, the piezoelectric actuators were slow and produced a high-pitched sound. We would suggest to any researchers wanting to replicate our piezoelectric automated stage scanning set-up to consider purchasing a different motion control actuator. Our script could be incorporated into any XYZ motion control system by simply substituting the motion control command lines with an alternative command. This script could also be adapted for time lapse imaging of multiple regions within a sample or for automated behaviour tracking.

# 5. Conclusion

The configurations described here are only three of the many potential configurations of the Flexiscope. Our system could be directly replicated by the reader or this study could be used as inspiration for any research group to establish their own custom Flexiscope. Depending on the needs and financial resources of the user, a stripped-back version or an upgraded model could be implemented. For example, if the user only requires one channel of fluorescence, a considerable reduction in cost would be achieved. The modular nature of the system means that as technologies improve, or as additional funding becomes available, new components can be incorporated or replaced over time as the needs of the team evolve. Innovation and flexibility in experimental design are paramount to the advancement of neuroscience.

Ethics. All work was performed in accordance with local institute guidelines and relevant Irish and EU legislation (directive 2010/63/EU and SI No. 543/2012). The experiments using DRG neuron cultures involved the use of rat pups and were performed with approval for this work granted by the Animal Research Ethics Committee, University College Dublin (AREC-16-04-Pickering). Work using ctenophore tissue was formally exempted from full ethical review (AREC-E-17-22-Pickering) on the grounds of being derived from non-protected invertebrate animals, again in accordance with both relevant legislation and institutional guidelines.

Data accessibility. Our data are available in electronic supplementary material, file S2.

Authors' contributions. A.C. and M.P. designed and constructed the Flexiscope. A.C. wrote the automation code, collected and analysed ctenophore imaging data and drafted the manuscript. L.M.A. carried out the DRG culture preparation and imaging. G.O.T.M. assisted in the optimization of the Flexiscope design. N.B. performed and analysed the resolution and mechanical stability tests. M.P. devised the study and oversaw its execution and edited the manuscript. All authors gave final approval for publication.

Competing interests. The authors declare no competing interests

Funding. This work was supported in full by School of Medicine, University College Dublin.

Acknowledgements. The authors wish to acknowledge Dominic Courtney for his invaluable assistance in the collection of ctenophores.

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
