## [Reviewer comments · Royal Society Open Science]

Review History

RSOS-191949.R0 (Original submission)

Review form: Reviewer 1 (Ricardo Henriques)

Is the manuscript scientifically sound in its present form?

Yes

Are the interpretations and conclusions justified by the results?

Yes

Is the language acceptable?

Yes

Do you have any ethical concerns with this paper?

No

Have you any concerns about statistical analyses in this paper?

Yes

Recommendation?

Accept with minor revision (please list in comments)

Comments to the Author(s)

Courtney et al. present a well-documented and thoroughly characterised microscope set-up based on common optomechanics that can be obtained from Thorlabs and through 3D printing. The set-up affords flexibility for upright, inverted microscopy and micromanipulation system, common for neuroscience studies. The manuscript is well written and cites a large section of the open-source microscopy literature. There is indeed already a number of designs in the literature on cheap/3D printed/hacked systems, well-aligned with the developing field of open science/microscopy. This work describes a combination of hardware that I have not seen anywhere else (including a comparison of piezo actuators and stepper motor-based translation stage) and develops the aspect of modularity nicely. The capabilities and the limitations of the piezo actuators are nicely characterised and in some cases solutions to mitigate these are proposed. I recommend the work for publication providing that the authors address the comments below. These are mainly about clarifications of a few details.

- The parts for the oblique IR illumination seem to be missing and the assembly is not as detailed as the rest of the system. What IR LEDs are used and how are they powered/controlled/mounted? Illumination from the side as is shown in Figure 4 typically leads to shadowing and diffraction patterns. Was this a problem in this case and if so, how was this solved?

- "Oblique infrared illumination microscopy (OIR) uses infrared LEDs (CO26) at an angle above or below the specimen which allows the visualisation of 3D structures, resulting in images similar in appearance to DIC microscopy 9 10." There is often a confusion in the field between DIC and bright-field illumination. I think the authors refer to bright-field illumination in this case. Darkfield could also be obtained with their setup (low NA and side-ways illumination), which could be useful here.

- As rightly and nicely discussed in the conclusion, MATLAB and its toolboxes are not an open-source platform and therefore do not strictly align with the open-source philosophy of this paper. So I think the price of the necessary MATLAB license and components should be explicitly added to the cost breakdown in Table 1. Also, Micro-manager should be mentioned/cited somewhere as a common microscopy hardware integration platform for completeness' sake.

- One major limitation of the piezo actuators described and characterised by the authors here is that they do not provide absolute displacement. This is discussed in 3.2.1 but the readability of the manuscript would improve by mentioning it and introducing how this was tackled earlier in the manuscript. Also, it should be made clear to the reader that the load/torque will influence the displacement of the piezo and therefore a calibration will be required for each mechanical configuration/sample mounting approach. Also, what are the theoretical limitations of these actuators in terms of resolution and range?

- I am worried about the mechanical stability of such systems in general, this is commonly a problem for custom-built systems like this one, especially the micromanipulator configuration with the whole microscope on the translation system. Have the authors looked at drift (xyz) over time of the systems? This can be done with looking at large and immobile beads for instance. This will of course mainly matter in the case of time-course measurements which the authors do not intend the system for but can be useful for versatility.

- On the topic of resolution (page 17), the authors look at USAF target. This is great but a quick comparison to Abbe/Rayleigh resolution estimation may also be useful to give an estimation of the resolutions achievable.

- The authors mention that it takes 290s to change FOV when using the piezo actuators, can the authors discuss why it takes so long?

- Regarding the stepper motor implementation, more details on how the stepper motor was coupled to the translation are needed. Use of gears? What is the range and resolution? How much does one step represent in physical sample space? The intention of such papers is for other labs to replicate the system and these pieces of information would help. I, for one, would be interested in replicating it.

Typos:

“We also characterised to stability of this stage during motion.”

“The X and Y-actuators where tested and optimised for 20X and 4X objectives to determine the appropriate travel distance to achieve a new ‘FOV’ with sufficient overlap for subsequent image stitching.”

Review form: Reviewer 2 (Richard Bowman)

Is the manuscript scientifically sound in its present form?

Yes

Are the interpretations and conclusions justified by the results?

Yes

Is the language acceptable?

Yes

Do you have any ethical concerns with this paper?

No

Have you any concerns about statistical analyses in this paper?

No

Recommendation?

Accept with minor revision (please list in comments)

Comments to the Author(s)

The manuscript describes a modular design for a microscope suitable for brightfield and epifluorescence imaging, constructed from industry-standard ThorLabs optomechanical components. The design looks to be very competently put together, well explained, and versatile enough to be convertible between upright, inverted, and electrophysiology configurations. Given that this conversion takes ~half an hour (presumably by someone who knows what they are doing already), I can imagine that this is rather less than perfectly convenient - but there are plenty circumstances in which a little convenience must be sacrificed for a significant cost saving. Practical solutions for labs where an extra £20k simply isn't available are a valuable contribution to the community, and so I'm glad to see this paper.

The authors have validated the imaging with a number of experiments using fluorescence microscopy, and it looks like it all works quite nicely. I am particularly pleased to see their data quantifying the uniformity of fluorescence illumination, as this is the sort of thing that is often missing from papers on DIY instruments.

I have a few queries and suggestions that I think might help the authors improve the manuscript, but I think these are all relatively minor.

Firstly, on the issue of converting between upright/inverted/electrophysiology configurations, I wonder if there are measures that might, at moderate cost, make the conversion substantially easier and quicker. For example, spending a few hundred points on some magnetic kinematic mounts and extra posts/platforms might cut the time to switch between configurations substantially, and ensure each configuration is precisely the same each time. For an even more

convenient system, if one were to duplicate some of the translation stages, would it be possible to move just the optics between three different mechanical mounts? Every lab will have their own tradeoff between convenience and cost, but I could see a small increase in cost having a potentially very large pay-off in convenience.

Secondly, while I'm a keen advocate of open and DIY hardware, I do always worry a little about the hidden cost of assembly time (which is often carried out by students who would otherwise be focused on a different piece of science). I think in the discussion of cost, it is important to quantify the time required (ideally including sourcing components, etc. and allowing for someone who has not built one before). Accounting for the additional staff time involved, systems such as the Cerna modular microscope are not so far removed from the price point of the Flexiscope - though of course it is often preferable, or simpler to arrange, to build a system in-house - particularly if it is necessary to have the expertise to customise it later.

The characterisation of the optical performance using a USAF target is a very sensible choice, though of course (as the authors point out) the lines on the target are sufficiently far apart that simply resolving it does not provide a very rigorous test of a microscope's resolution. The line profiles included are helpful, and could in principle allow the resolution to be quantified, but one must be very careful at that point to avoid saturation in the images - I've not inspected the raw data but the images included in the manuscript do have very white backgrounds. I would suggest that a good way to quantify the resolution would be using the slant-edge MTF method, which can be done using one of the large opaque squares on the same USAF target slide that is imaged in the manuscript. If this is not a significant amount of extra work, it might be nice to see - but as the optical design is quite straightforward, I would expect that its performance is mostly limited by the objective, at least in the centre of the field of view.

Finally, having looked through the instruction manual, I think it is currently good enough for someone familiar with the construction of optical systems to reproduce precisely what the authors have built. However, if this is to be accessible to non-specialists, my experience of sharing instrument designs leads me to think that the instructions will need to be considerably more detailed. While I would love to make this level of documentation a requirement for published instruments, that is not yet the accepted standard for the vast majority of journals, and so it should not hold up publication. I will, however, encourage the authors to consider improving their documentation if they hope to see the system implemented more widely. Good documentation that is already available will attract people to the project, and many potential users will not get in touch if the project isn't obviously documented for the non-specialist - so leaving more detailed instructions until later is not always an effective strategy.

Other than that, I enjoyed reading the manuscript, and I hope that this proves a useful resource for others who need to construct a similar system.

Decision letter (RSOS-191949.R0)

08-Jan-2020

Dear Ms Courtney

On behalf of the Editors, I am pleased to inform you that your Manuscript RSOS-191949 entitled "The Flexiscope: a Low Cost, Flexible, Convertible, and Modular Microscope with Automated Scanning and Micromanipulation." has been accepted for publication in Royal Society Open Science subject to minor revision in accordance with the referee suggestions. Please find the referees' comments at the end of this email.

The reviewers and handling editors have recommended publication, but also suggest some minor revisions to your manuscript. Therefore, I invite you to respond to the comments and revise your manuscript.

- Ethics statement

- Data accessibility

If you wish to submit your supporting data or code to Dryad (<http://datadryad.org/>), or modify your current submission to dryad, please use the following link:
<http://datadryad.org/submit?journalID=RSOS&manu=RSOS-191949>

- Competing interests

- Authors' contributions

- Acknowledgements

- Funding statement

Please ensure you have prepared your revision in accordance with the guidance at <https://royalsociety.org/journals/authors/author-guidelines/> -- please note that we cannot

publish your manuscript without the end statements. We have included a screenshot example of the end statements for reference. If you feel that a given heading is not relevant to your paper, please nevertheless include the heading and explicitly state that it is not relevant to your work.

Because the schedule for publication is very tight, it is a condition of publication that you submit the revised version of your manuscript before 17-Jan-2020. Please note that the revision deadline will expire at 00.00am on this date. If you do not think you will be able to meet this date please let me know immediately.

If your manuscript is newly submitted and subsequently accepted for publication, you will be asked to pay the article processing charge, unless you request a waiver and this is approved by Royal Society Publishing. You can find out more about the charges at <https://royalsocietypublishing.org/rsos/charges>. Should you have any queries, please contact openscience@royalsociety.org.

on behalf of Dr Peter Munro (Associate Editor) and Kevin Padian (Subject Editor)
openscience@royalsociety.org

Editor comments:

Thanks for your submission, which as you can see the reviewers very much liked. They raise a number of suggestions that should not be difficult to address and should improve the manuscript for readers. If you have trouble making these revisions within the allotted time please contact our editorial office. Best wishes.

Reviewer comments to Author:

Reviewer: 1

Comments to the Author(s)

Courtney et al. present a well-documented and thoroughly characterised microscope set-up based on common optomechanics that can be obtained from Thorlabs and through 3D printing. The set-up affords flexibility for upright, inverted microscopy and micromanipulation system, common for neuroscience studies. The manuscript is well written and cites a large section of the open-source microscopy literature. There is indeed already a number of designs in the literature on cheap/3D printed/hacked systems, well-aligned with the developing field of open science/microscopy. This work describes a combination of hardware that I have not seen anywhere else (including a comparison of piezo actuators and stepper motor-based translation stage) and develops the aspect of modularity nicely. The capabilities and the limitations of the piezo actuators are nicely characterised and in some cases solutions to mitigate these are proposed. I recommend the work for publication providing that the authors address the comments below. These are mainly about clarifications of a few details.

- The parts for the oblique IR illumination seem to be missing and the assembly is not as detailed as the rest of the system. What IR LEDs are used and how are they powered/controlled/mounted? Illumination from the side as is shown in Figure 4 typically leads to shadowing and diffraction patterns. Was this a problem in this case and if so, how was this solved?

- "Oblique infrared illumination microscopy (OIR) uses infrared LEDs (CO26) at an angle above or below the specimen which allows the visualisation of 3D structures, resulting in images similar in appearance to DIC microscopy 9 10." There is often a confusion in the field between DIC and bright-field illumination. I think the authors refer to bright-field illumination in this case. Darkfield could also be obtained with their setup (low NA and side-ways illumination), which could be useful here.

- As rightly and nicely discussed in the conclusion, MATLAB and its toolboxes are not an open-source platform and therefore do not strictly align with the open-source philosophy of this paper. So I think the price of the necessary MATLAB license and components should be explicitly added to the cost breakdown in Table 1. Also, Micro-manager should be mentioned/cited somewhere as a common microscopy hardware integration platform for completeness' sake.

- One major limitation of the piezo actuators described and characterised by the authors here is that they do not provide absolute displacement. This is discussed in 3.2.1 but the readability of the manuscript would improve by mentioning it and introducing how this was tackled earlier in the manuscript. Also, it should be made clear to the reader that the load/torque will influence the displacement of the piezo and therefore a calibration will be required for each mechanical configuration/sample mounting approach. Also, what are the theoretical limitations of these actuators in terms of resolution and range?

- I am worried about the mechanical stability of such systems in general, this is commonly a problem for custom-built systems like this one, especially the micromanipulator configuration with the whole microscope on the translation system. Have the authors looked at drift (xyz) over time of the systems? This can be done with looking at large and immobile beads for instance. This will of course mainly matter in the case of time-course measurements which the authors do not intend the system for but can be useful for versatility.

- On the topic of resolution (page 17), the authors look at USAF target. This is great but a quick comparison to Abbe/Rayleigh resolution estimation may also be useful to give an estimation of the resolutions achievable.

- The authors mention that it takes 290s to change FOV when using the piezo actuators, can the authors discuss why it takes so long?

- Regarding the stepper motor implementation, more details on how the stepper motor was coupled to the translation are needed. Use of gears? What is the range and resolution? How much does one step represent in physical sample space? The intention of such papers is for other labs to replicate the system and these pieces of information would help. I, for one, would be interested in replicating it.

Typos:

“We also characterised to stability of this stage during motion.”

“The X and Y-actuators where tested and optimised for 20X and 4X objectives to determine the appropriate travel distance to achieve a new ‘FOV’ with sufficient overlap for subsequent image stitching.”

Reviewer: 2

Comments to the Author(s)

The manuscript describes a modular design for a microscope suitable for brightfield and epifluorescence imaging, constructed from industry-standard ThorLabs optomechanical components. The design looks to be very competently put together, well explained, and versatile enough to be convertible between upright, inverted, and electrophysiology configurations. Given that this conversion takes ~half an hour (presumably by someone who knows what they are doing already), I can imagine that this is rather less than perfectly convenient - but there are plenty circumstances in which a little convenience must be sacrificed for a significant cost saving. Practical solutions for labs where and extra £20k simply isn't available are a valuable contribution to the community, and so I'm glad to see this paper.

The authors have validated the imaging with a number of experiments using fluorescence microscopy, and it looks like it all works quite nicely. I am particularly pleased to see their data quantifying the uniformity of fluorescence illumination, as this is the sort of thing that is often missing from papers on DIY instruments.

I have a few queries and suggestions that I think might help the authors improve the manuscript, but I think these are all relatively minor.

Firstly, on the issue of converting between upright/inverted/electrophysiology configurations, I wonder if there are measures that might, at moderate cost, make the conversion substantially easier and quicker. For example, spending a few hundred points on some magnetic kinematic mounts and extra posts/platforms might cut the time to switch between configurations substantially, and ensure each configuration is precisely the same each time. For an even more convenient system, if one were to duplicate some of the translation stages, would it be possible to move just the optics between three different mechanical mounts? Every lab will have their own tradeoff between convenience and cost, but I could see a small increase in cost having a potentially very large pay-off in convenience.

Secondly, while I'm a keen advocate of open and DIY hardware, I do always worry a little about the hidden cost of assembly time (which is often carried out by students who would otherwise be focused on a different piece of science). I think in the discussion of cost, it is important to quantify the time required (ideally including sourcing components, etc. and allowing for someone who has not built one before). Accounting for the additional staff time involved, systems such as the Cerna modular microscope are not so far removed from the price point of the Flexiscope - though of course it is often preferable, or simpler to arrange, to build a system in-house - particularly if it is necessary to have the expertise to customise it later.

The characterisation of the optical performance using a USAF target is a very sensible choice, though of course (as the authors point out) the lines on the target are sufficiently far apart that simply resolving it does not provide a very rigorous test of a microscope's resolution. The line profiles included are helpful, and could in principle allow the resolution to be quantified, but one must be very careful at that point to avoid saturation in the images - I've not inspected the raw data but the images included in the manuscript do have very white backgrounds. I would suggest that a good way to quantify the resolution would be using the slant-edge MTF method, which can be done using one of the large opaque squares on the same USAF target slide that is imaged in the manuscript. If this is not a significant amount of extra work, it might be nice to see - but as the optical design is quite straightforward, I would expect that its performance is mostly limited by the objective, at least in the centre of the field of view.

Finally, having looked through the instruction manual, I think it is currently good enough for someone familiar with the construction of optical systems to reproduce precisely what the authors have built. However, if this is to be accessible to non-specialists, my experience of sharing instrument designs leads me to think that the instructions will need to be considerably more detailed. While I would love to make this level of documentation a requirement for published instruments, that is not yet the accepted standard for the vast majority of journals, and so it should not hold up publication. I will, however, encourage the authors to consider improving their documentation if they hope to see the system implemented more widely. Good documentation that is already available will attract people to the project, and many potential users will not get in touch if the project isn't obviously documented for the non-specialist - so leaving more detailed instructions until later is not always an effective strategy.

Other than that, I enjoyed reading the manuscript, and I hope that this proves a useful resource for others who need to construct a similar system.

Author's Response to Decision Letter for (RSOS-191949.R0)

See Appendix A.

Decision letter (RSOS-191949.R1)

07-Feb-2020

Dear Ms Courtney,

It is a pleasure to accept your manuscript entitled "The Flexiscope: a Low Cost, Flexible, Convertible, and Modular Microscope with Automated Scanning and Micromanipulation." in its current form for publication in Royal Society Open Science. The comments of the reviewer(s) who reviewed your manuscript are included at the foot of this letter.

on behalf of Dr Peter Munro (Associate Editor) and Kevin Padian (Subject Editor)
openscience@royalsociety.org

Appendix A

First, we would like to thank the reviewers for taking the time to review our manuscript, and for their thoughtful comments which we believe have improved our manuscript. We have included our response to their specific comments below.

Reviewer: 1

Comments to the Author(s)

“The parts for the oblique IR illumination seem to be missing and the assembly is not as detailed as the rest of the system. What IR LEDs are used and how are they powered/controlled/mounted? Illumination from the side as is shown in Figure 4 typically leads to shadowing and diffraction patterns. Was this a problem in this case and if so, how was this solved?”

The IR illuminator we used is a ring type 36 LED array commonly used in CCTV systems, where the ring of LEDs surround the CCTV camera lens. These arrays have integrated LED drivers, so simply need to be provided with 12v DC power to operate. The array is attached to adjustable angle posts (parts MO6 and MO7) which allow the angle of illumination to be adjusted for desired results. We didn't notice any problems with shadowing or diffraction (as seen in figure 4), but this may be due to the wide and relatively diffuse illumination produced by the array (as opposed to single LEDs). The text in the revised manuscript now includes these details in the description (page 5 and 6).

The specific IR array we used was salvaged from a decommissioned CCTV camera. However, these IR illuminators are common and easy to find, but we have found a source for a similar illuminator (Amazon) and updated the entry on the parts list to include this.

- “Oblique infrared illumination microscopy (OIR) uses infrared LEDs (CO26) at an angle above or below the specimen which allows the visualisation of 3D structures, resulting in images similar in appearance to DIC microscopy 9 10.” There is often a confusion in the field between DIC and bright-field illumination. I think the authors refer to bright-field illumination in this case. Darkfield could also be obtained with their setup (low NA and side-ways illumination), which could be useful here.

We did intend to draw the comparison to DIC, although we should have been clearer about the nature of this comparison. The use of OIR in our design was largely inspired by the use of OIR in microscopy set-ups designed for electrophysiological studies. Infrared DIC has been a dominant technique in experiments requiring the positioning of an electrode onto a single cell within a relatively thick (150µm and upwards) tissue sample. However, as DIC requires specialised optical elements in both the illumination and detection light path, others have sought to replace it with simpler, more accessible alternatives. Previous studies have explicitly made the comparison of DIC to OIR illumination, which is why the previous version of the manuscript mentioned the similarity of both. However, we can see that we didn't make it clear that the comparison between the two techniques was not ours, but had been carried out previously.

We have revised the manuscript (page 6) to clarify this point and refer to the published comparisons of OIR and DIC illumination. We have also changed reference 10 to a more appropriate reference; a paper which explicitly compares DIC and OIR microscopy.

“As rightly and nicely discussed in the conclusion, MATLAB and its toolboxes are not an open-source platform and therefore do not strictly align with the open-source philosophy of this paper. So I think the price of the necessary MATLAB license and components should be explicitly added to the cost breakdown in Table 1. Also, Micro-manager should be mentioned/cited somewhere as a common microscopy hardware integration platform for completeness' sake.”

The price of Matlab has been included in the table and the discussion (page 18). We have also included a discussion on μ Manager integration at the end of section 4.2 (page 18)

“One major limitation of the piezo actuators described and characterised by the authors here is that they do not provide absolute displacement. This is discussed in 3.2.1 but the readability of the manuscript would improve by mentioning it and introducing how this was tackled earlier in the manuscript. Also, it should be made clear to the reader that the load/torque will influence the displacement of the piezo and therefore a calibration will be required for each mechanical configuration/sample mounting approach. Also, what are the theoretical limitations of these actuators in terms of resolution and range?”

We have included discussion of the issues associated with the piezo system earlier in the manuscript, in the methods section where we describe the piezo system and explain why the stepper system might be needed (section 2.2, page 7-8). This section now also includes the resolution and travel range of these actuators.

“I am worried about the mechanical stability of such systems in general, this is commonly a problem for custom-built systems like this one, especially the micromanipulator configuration with the whole microscope on the translation system. Have the authors looked at drift (xyz) over time of the systems? This can be done with looking at large and immobile beads for instance. This will of course mainly matter in the case of time-course measurements which the authors do not intend the system for but can be useful for versatility.”

The stability of the system is shown indirectly in our assessment of the stability of the microelectrode position, which is documented in section 3.3 (page 16). Over 16 hours, using the 4x objective, we saw no movement of the electrode tip, which implies that not only is the electrode stable, but so is the microscope.

However, as that experiment was performed with relatively low resolution, we have performed an additional measurement of stability. We imaged the corner of a solid square on the USAF grid in over a period of 4 hours at 10 minute intervals, and used the movement of the location of the corner pixel to estimate drift. This is now documented on pages 11 and 12 and in supplementary figure 3. This worked well for X and Y axis, but we were not able to accurately measure Z drift on our system. Z drift would result in focus shift, and, we observed no evidence of focus shift over these 4 hours.

- On the topic of resolution (page 17), the authors look at USAF target. This is great but a quick comparison to Abbe/Rayleigh resolution estimation may also be useful to give an estimation of the resolutions achievable.

We have included an estimate of the theoretical resolution limit on page 11.

On the recommendation of reviewer two, we also evaluated the capability of the system using the slant edge MTF approach. This is detailed on pages 11 and 12, with the results detailed in supplementary figure 3

“The authors mention that it takes 290s to change FOV when using the piezo actuators, can the authors discuss why it takes so long?”

The maximum speed is defined by the step size, and the maximum step frequency. This means piezo actuators with small step sizes are always going to be slow. Thorlabs state a maximum speed of 3.5mm/minute, or 2mm/minute typical, for these piezo actuators. The combination of driver and actuator we used was not capable of these speeds, which we suspect is due to a combination the driver type we used, and the load on the actuator. We have included a brief discussion of this point on page 15.

One key difference between stepper and piezo systems is that steppers can be driven in full steps or microsteps, allowing them to be both fast and precise. This is related to the next point raised.

“Regarding the stepper motor implementation, more details on how the stepper motor was coupled to the translation are needed. Use of gears? What is the range and resolution? How much does one step represent in physical sample space? The intention of such papers is for other labs to replicate the system and these pieces of information would help. I, for one, would be interested in replicating it.”

A more detailed description of the stepper system has been included in section 2.2, page 7 and 8. Steppers are coupled directly to the micrometers so the rotation of the stepper is translated into rotation of the micrometer. The steppers are coupled to 13mm travel micrometers that translate 500 μ m/revolution. Our stepper operates at 200 steps/revolution, which gives 2.5 μ m/step. However, steppers can be driven at fractions of steps by microstepping. Using 1/16 microstepping allows a theoretical limit of 156.25nm/microstep.

Typos:

“We also characterised to stability of this stage during motion.”

“The X and Y-actuators were tested and optimised for 20X and 4X objectives to determine the appropriate travel distance to achieve a new ‘FOV’ with sufficient overlap for subsequent image stitching.”

These typos have been fixed in the revised manuscript.

Reviewer: 2

“Firstly, on the issue of converting between upright/inverted/electrophysiology configurations, I wonder if there are measures that might, at moderate cost, make the conversion substantially easier and quicker. For example, spending a few hundred points on some magnetic kinematic mounts and extra posts/platforms might cut the time to switch between configurations substantially, and ensure each configuration is precisely the same each time. For an even more convenient system, if one were to duplicate some of the translation stages, would it be possible to move just the optics between three different mechanical mounts? Every lab will have their own tradeoff between convenience and cost, but I could see a small increase in cost having a potentially very large pay-off in convenience.”

This is a very interesting idea, and one we considered early in the design of the system. However because we found that, in our own use of the system, conversions between configurations tend to be relatively rare (less than once/month), the potential drawbacks of magnetic kinematic mounts (potential loss on stability or rigidity, as well as cost) outweighed the advantage in time savings, we chose not to pursue this avenue in our design. I also don't believe all parts of the system as described would be achievable with kinematic mounts. For example, in some configurations (specifically the electrophysiology configuration) the entire mass of the optical system is mounted

horizontally, and I am not aware of a system of kinematic mounts that would permit this without a significant loss of stability.

That said, the system we present here can be seen as an example of a design that can be adapted in many ways depending on the users needs. There is no reason why others couldn't adapt our design in the way you describe, if that suited their needs.

"Secondly, while I'm a keen advocate of open and DIY hardware, I do always worry a little about the hidden cost of assembly time (which is often carried out by students who would otherwise be focused on a different piece of science). I think in the discussion of cost, it is important to quantify the time required (ideally including sourcing components, etc. and allowing for someone who has not built one before). Accounting for the additional staff time involved, systems such as the Cerna modular microscope are not so far removed from the price point of the Flexiscope - though of course it is often preferable, or simpler to arrange, to build a system in-house - particularly if it is necessary to have the expertise to customise it later."

This, again, is a good point to raise. We did invest a significant amount of time during the design stage experimenting with different setups and configurations. However, now that we have established the design, assembly is relatively quick and straightforward. Because the system is optically simple, there are no difficult alignment steps which can take considerable time.

We believe, with all components to hand, even a complete novice user should be able to assemble the system within a day, and someone with some experience would be considerably quicker. All components should be easily available, so sourcing components shouldn't take too much time.

We have added a statement about assembly times in the discussion in the revised manuscript in the first paragraph of the discussion (page 16 and 17)

"The characterisation of the optical performance using a USAF target is a very sensible choice, though of course (as the authors point out) the lines on the target are sufficiently far apart that simply resolving it does not provide a very rigorous test of a microscope's resolution. The line profiles included are helpful, and could in principle allow the resolution to be quantified, but one must be very careful at that point to avoid saturation in the images - I've not inspected the raw data but the images included in the manuscript do have very white backgrounds. I would suggest that a good way to quantify the resolution would be using the slant-edge MTF method, which can be done using one of the large opaque squares on the same USAF target slide that is imaged in the manuscript. If this is not a significant amount of extra work, it might be nice to see - but as the optical design is quite straightforward, I would expect that its performance is mostly limited by the objective, at least in the centre of the field of view."

We'd like to thank the reviewer for this excellent suggestion. I particularly like that this method measures the resolving capabilities of the system as a whole, including the properties of the optics and the sensor, which is especially important when we are using a camera system that is not specifically designed for microscopy. We have included our slant edge MTF measurements on page 11 and 12 and in supplementary figure 3

"Finally, having looked through the instruction manual, I think it is currently good enough for someone familiar with the construction of optical systems to reproduce precisely what the authors have built. However, if this is to be accessible to non-specialists, my experience of sharing instrument designs leads me to think that the instructions will need to be considerably more detailed. While I would love to make this level of documentation a requirement for published instruments, that is not

yet the accepted standard for the vast majority of journals, and so it should not hold up publication. I will, however, encourage the authors to consider improving their documentation if they hope to see the system implemented more widely. Good documentation that is already available will attract people to the project, and many potential users will not get in touch if the project isn't obviously documented for the non-specialist - so leaving more detailed instructions until later is not always an effective strategy."

This raises a very good point about quality and clarity of documentation, and the type of documentation that different users would require. The key idea of the assembly manual was to highlight crucial steps in the correct assembly of critical components, which we hope this achieves. However, we agree that a more detailed, step by step instruction set, with extensive use of photos/diagrams and perhaps videos, would also be of use to many potential users. However, we believe this type of documentation works best as a live, dynamic format. One of the best examples of assembly documents I have come across are those used by Prusa Research for their 3D printers. However, this quality is the result of continuous revision and updates based on user feedback. I think a separate, post-publication detailed assembly instruction would be appropriate in this case, as it would allow us to revise it continuously based on feedback. Also, using an open resource such as github for this would allow us and others to update the information with modifications, additions, and revisions to the system itself. Our microscope design has evolved as we worked with it, and we would envision the design evolving into the future as we (and hopefully others) continue to use it.

Additional changes in revised manuscript.

The closing statements have been revised in line with the journal guidelines.

An additional author has been added (Niamh Burke)- this author carried out the stability and resolution testing included in the revised manuscript.